# Patterns of Circulating Cytokines and Vascular Markers’ Response in the Presence of COVID-19 in Kidney Transplant Recipients Compared with Non-Transplanted Patients

**DOI:** 10.3390/v15112166

**Published:** 2023-10-28

**Authors:** Milena Karina Coló Brunialti, Giuseppe G. F. Leite, Gabriela Strafolino Eburneo, Orlei Ribeiro de Araujo, Paula M. Peçanha-Pietrobom, Paulo Roberto Abrão Ferreira, Nancy C. Junqueira Bellei, Jaquelina Sonoe Ota Arakaki, José Medina-Pestana, Lúcio Requião-Moura, Reinaldo Salomao

**Affiliations:** 1Division of Infectious Diseases, Escola Paulista de Medicina, Universidade Federal de São Paulo, São Paulo 04023-900, Brazil; brmilenabr@gmail.com (M.K.C.B.); giuseppe.gianini@unifesp.br (G.G.F.L.); gabistrafolino@gmail.com (G.S.E.); paulampecanha@gmail.com (P.M.P.-P.); paulo.abrao.ferreira@gmail.com (P.R.A.F.); nbellei@uol.com.br (N.C.J.B.); 2Intensive Care Unit, GRAACC, Pediatric Institute of Oncology, Universidade Federal de São Paulo, São Paulo 04039-001, Brazil; orlei@uol.com.br; 3Division of Respiratory Diseases, Escola Paulista de Medicina, Universidade Federal de São Paulo, São Paulo 04020-050, Brazil; jaqueota@gmail.com; 4Division of Nephrology, Universidade Federal de São Paulo, São Paulo 04038-031, Brazil; medina@hrim.com.br; 5Hospital do Rim, Fundação Oswalado Ramos, São Paulo 04038-002, Brazil; 6Hospital São Paulo, São Paulo 04024-002, Brazil

**Keywords:** SARS-CoV-2, kidney transplant recipients, cytokines, vascular mediators, correlation-based network

## Abstract

COVID-19’s severity has been associated with a possible imbalance in the cross-regulation of cytokines and vascular mediators. Since the beginning of the pandemic, kidney transplant recipients (KTRs) have been identified as patients of high vulnerability to more severe diseases. Thus, aiming to describe the patterns of cytokines and vascular mediators and to trace patients’ differences according to their KTR status, this prospective study enrolled 67 COVID-19 patients (20 KTRs) and 29 non-COVID-19 controls before vaccination. A panel comprising 17 circulating cytokines and vascular mediators was run on samples collected at different time points. The cytokine and mediator patterns were investigated via principal component analysis (PCA) and correlation-based network (CBN). In both groups, compared to their respective controls, COVID-19 was associated with higher levels of cytokines and vascular mediators. Differentiating between the KTRs and non-KTRs, the number of correlations was much higher in the non-KTRs (44 vs. 14), and the node analysis showed the highest interactions of NGAL and sVCAM-1 in the non-KTRs and KTRs (9 vs. 4), respectively. In the PCA, while the non-KTRs with COVID-19 were differentiated from their controls in their IL-10, IFN-α, and TNF-α, this pattern was marked in the NGAL, sVCAM-1, and IL-8 of the KTRs.

## 1. Introduction

The widespread vaccination against SARS-CoV-2 reduced the severity and the number of cases in countries with high vaccination coverage rates, consequently changing the scenario of the COVID-19 pandemic [1]. As a result, the disease seems to be under control, which recently motivated the WHO to state that COVID-19 no longer constitutes a public health emergency of international concern [2]. However, since some populational clusters did not present an effective response to vaccination, they still suffered the severe effects of the infection, such as the recipients of solid organ transplantation [3,4,5]. In an analysis based on the age-adjusted case fatality rates among kidney transplant recipients (KTRs) compared with the overall population in Brazil, the risk of death was more than seven-fold higher for KTRs, and for them, the age-adjusted risk of death did not change with vaccination up to the Omicron prevalence [6]. Furthermore, these findings were confirmed in studies conducted in other countries [4], underscoring the high vulnerability among these patients even after vaccination.

The clinical presentations and outcomes in KTRs awakened particular concerns, due to their unavoidable lifelong immunosuppression maintenance and their cumulative number of comorbidities [7,8], which was demonstrated in different cohorts [9,10]. Despite these impacting results, the effect of the presence of the kidney graft per se and the chronic use of immunosuppressive agents on the imbalance between circulating cytokines, microangiopathy, thrombosis and, consequently, on the inflammatory response, and on endothelial dysfunction is not known and has not been previously investigated.

In individuals with primary or secondary immunity dysfunction, the interaction of SARS-CoV-2 with target cells, the increased circulating cytokine levels, and the imbalance between pro- and anti-inflammatory mediators seem to be a cornerstone of COVID-19’s severity [11,12,13]. Therefore, increased plasma levels of cytokines, referred to as a cytokine storm, have been considered a pivotal event in COVID-19′s disease pathophysiology and a target for therapy [14,15]. Beyond cytokine storms, microangiopathy, endothelial dysfunction, and widespread thrombosis were also observed in patients who deteriorated over the course of the disease [16,17,18]. Nevertheless, the similarities and differences between sepsis and COVID-19 regarding cytokine-induced disturbances in coagulation [19,20,21] and thrombosis mechanisms were demonstrated [22,23]. 

We prospectively followed a cohort of COVID-19 patients evaluated before vaccination had been started in Brazil, focusing on those affected by a moderate to severe disease who were admitted into the wards of a tertiary hospital, including KTRs. The clinical and epidemiological characteristics and routine laboratory changes were previously published [24]. Herein, we evaluated the plasma levels of cytokines and vascular mediators assessed at hospital admission and at different time points over the course of the disease, aiming to describe their signature and to trace their possible differences according to their kidney transplant status.

## 2. Methods

### 2.1. Design Study, Population, and Setting

This prospective cohort study recruited individuals aged over 18 and diagnosed with COVID-19 who were admitted into Hospital São Paulo’s wards, the Federal University of São Paulo, Brazil between May and September 2020. Their COVID-19 diagnoses were confirmed via polymerase chain reaction (PCR) testing using nasopharyngeal swabs. The patients received standard hospital protocol-based treatment.

A total of 67 COVID-19 patients with moderate-to-severe disease, defined according to the World Health Organization (WHO) and National Institutes of Health (NIH) guidelines, were enrolled, including 20 KTRs [25,26]. The control group consisted of 19 healthy individuals without any COVID-19 symptoms and with confirmed negative results via both serological and PCR tests. They were matched based on sex and age to maintain comparability. Additionally, another control group included 10 KTRs devoid of COVID-19 symptoms and any contact with the virus. This group was further matched concerning time since transplantation, immunosuppressive therapy, sex, and age. The study design is depicted in Figure 1.

### 2.2. Outcomes

The outcomes were progression to critical illness and mortality.

### 2.3. Blood Sampling and Plasma Detection of Circulating Mediators

Blood samples were acquired at four distinct time points during the study. Specifically, samples were obtained upon admission (D0, N = 67), on the 3rd day (D3, N = 50), on the 7th day (D7, N = 36), and upon post-hospital discharge from 43 survivors, with a median time of 33 days (referred to as CS30 in the results section). Samples were transported at 4 °C to the research laboratory, were processed within two hours, were frozen at −80 °C after centrifugation, and were kept until use. The details of the mediators and methodological specificity are shown in Table 1 and Appendix A.

### 2.4. Statistical Analysis

Statistical analysis was conducted using R version 4.2.1. To assess normality, the Shapiro–Wilk test was employed. The variables were expressed as median values accompanied by the 25th and 75th percentiles. Comparisons between groups were performed using either the Mann–Whitney U test or the Kruskal–Wallis test, followed by Dunn’s test, with Benjamini–Hochberg correction for multiple comparisons. Additionally, for categorical data, the chi-squared test was utilized for comparison. Statistically significant findings were denoted by the following symbols: * *p* < 0.05, ** *p* < 0.01, *** *p* < 0.001, and **** *p* < 0.0001.

### 2.5. Principal Component Analysis

PCA was performed using plasma biomarker abundances. The samples were grouped based on KTR status and the four time points. The respective controls were included in each analysis. R version 4.2.1 was utilized for PCA, specifically using the “prcomp” function. PCA plots were generated with the “fviz_pca_ind” function of the “factoextra” package to visualize group dispersion, and ellipses were added, representing 33% confidence intervals for each group and helping to identify general trends in the dataset. Furthermore, the “fviz_pca_var” function of the “factoextra” package was employed for the graphing of variables. We selected the top 3 contributing variables.

### 2.6. Correlation-Based Network (CBN) Analysis

Correlation networks were constructed by performing correlation analysis on all biomarkers. Spearman correlation coefficients (Rho) were calculated using the “*psych*” R package (version 2.3.3). Additionally, *p*-values were determined. Correlations meeting the criterion of *p*-value < 0.05 and a correlation coefficient of |rho| ≥ 0.20 were imported into Cytoscape (version 3.10.0) for network visualization [27].

## 3. Results

### 3.1. Baseline Characteristics and Outcomes

The kidney transplant recipients were 55.3 years old and 60% male, while the non-KTRs were 62.7 years old and 62.7% male, with no significant differences regarding age or sex distribution (Table 2). The most frequent maintenance immunosuppressive regimen was the combination of tacrolimus, mycophenolate, and prednisone in both the KTR groups, and all the regimens are detailed in Appendix A. Despite similar COVID-19-attributable symptoms between both groups, some other differences were observed in their baseline characteristics, as the KTRs had a higher frequency of CKD, higher initial creatinine and troponin levels, a higher SOFA score, a lower lymphocyte count, and a lower hemoglobin level. The length of hospital stay was higher among the KTRs (14.5 vs. 8.0 days), and they deteriorated more frequently to severe disease (50 vs. 28%), leading to a much higher mortality rate (40 vs. 8.5%).

### 3.2. Cytokines and Vascular Mediators Stratified by the Baseline Condition

As depicted in Figure 2, independent of the baseline clinical condition (KTR or not), COVID-19 impacted the cytokine landscape, with higher levels of cytokines and vascular mediators observed in the COVID-19 patients than in the healthy controls (HCs). These results are also detailed in Appendix A. At baseline, for non-KTRs, differences in several cytokines and vascular mediators were observed when the COVID-19 patients were compared with the HCs, except for ADAMTS-13, myoglobin, IL-9, G-CSF, and TNF-α. On day 3 after admission, significant differences were observed in GDF-15 (0.5 vs. 1.6 ng/mL), MPO (0.9 vs. 58.8 ng/mL), P-selectin (36.2 vs. 157.9 ng/mL), IL-6 (1.3 vs. 27.4 pg/mL), IL-8 (4.0 vs. 20.4 pg/mL), and IL-10 (0.0 vs. 3.7 pg/mL). While differences were sustained in GDF-15, P-selectin, and IL-10 on day 7 after admission, other mediators achieved significantly different levels, such as ADAMTS-13 (703.6 vs. 503.3 ng/mL), sICAM-1 (39.7 vs. 59.9 ng/mL), and SAA (1453.2 vs. 2437.7 ng/mL). 

The comparison between the KTRs with and without COVID-19 (the KTR controls) underscored a different pattern of progression, leastwise in terms of their cytokine and vascular mediator levels. At baseline, fewer differences were observed for KTRs than for non-KTRs, which were more evident with respect to MPO, sVCAM-1, SAA, IL-6, Il-8, Il-9, IL-10, and IFN-α. On day 3, differences were observed for MPO (12.9 vs. 78.4 ng/mL), IL-6 (5.3 vs. 29.1 pg/mL), IL-8 (9.6 vs. 21.9 pg/mL), IL-9 (0.0 vs. 1.6 pg/mL), IL-10 (1.8 vs. 7.5 pg/mL), and TNF-α (0.0 vs. 2.3 pg/mL). Lastly, only three markers were statistically different on day 7: MPO (12.9 vs. 78.7 ng/mL), IL-6 (5.3 vs. 46.5 pg/mL), and TNF-α (0.0 vs. 2.0 pg/mL).

Focusing on the analysis performed on the patients diagnosed with COVID-19, the comparison of both clinical situations (KTRs vs. non-KTRs) underscored a significant difference in INF-α (7.8 vs. 1.6 pg/mL) and NGAL (101.8 vs. 49.9 ng/mL) at baseline, a difference in myoglobin (62.9 vs. 29.6 ng/mL) and NGAL (58.0 vs. 39.5 ng/mL) on day 3, and no differences on day 7 after admission (Appendix A).

### 3.3. Correlation-Based Networks

To investigate the correlation between plasma biomarker levels early in the infection, four correlation-based networks (CBN) were constructed, as depicted in Figure 3. In all the groups, most of the significant correlations were positive, indicating that increases in the abundance of other biomarkers generally accompanied increases in the abundance of a specific biomarker. Remarkably, there was a significantly higher number of correlations observed in the non-KTR patients (44 interactions), compared to the KTRs (14 interactions). No negative correlations were detected in the KTR network. However, in the non-KTR network, two negative correlations were observed, specifically involving P-selectin and NGAL, as well as ADAMTS-15 and TNF-α. The node analysis of the non-KTR network revealed that NGAL and sVCAM-1 exhibited the highest number of interactions, with nine interactions each. These two markers also demonstrated the highest number of interactions in the KTR network, with four interactions each.

### 3.4. Cytokines and Vascular Mediators Stratified by Clinical Progression to Critical Illness

Considering their clinical progression, the patients were stratified according to severity, and the mediators in the KTRs and non-KTRs at the three time points were compared, as detailed in Appendix A. At baseline, myoglobin (140.6 vs. 40.7 ng/mL), sICAM-1 (124.2 vs. 65.6 ng/mL), NGAL (94.6 vs. 47.3 ng/mL), sVCAM-1 (2873 vs. 1646 ng/mL), IL-6 (112.9 vs. 45.4 pg/mL), IL-8 (40.6 vs. 24.9 pg/mL), IL-10 (11.0 vs. 5.3 pg/mL), and IFN-α (5.2 vs. 0.6 pg/mL) were significantly higher in the non-KTRs who progressed to critical condition than in those who presented with clinical recovery. Two other markers differentiated their clinical evolutions on day 3: GDF-15 (2.8 vs. 1.5 ng/mL) and P-selectin (113.6 vs. 210.9 ng/mL). On the other hand, among the KTRs, only IL-10 (12.2 vs. 6.1 pg/mL) was different in the patients who progressed to critical condition at baseline, and two other markers presented significant differences on day 3: myoglobin (114.2 vs. 33.1 ng/mL) and IL-8 (42.4 vs. 11.6 pg/mL). In addition to IL-8, IL-6 was also higher in the KTRs who progressed to critical illness on day 7 after admission (109.2 vs. 9.2 pg/mL).

### 3.5. Principal Component Analysis of Plasma Biomarkers

To identify the distinguishing features between the non-KTRs and KTRs at three different time points during hospitalization and 33 days after discharge, a principal component analysis (PCA) using plasma biomarkers was constructed (Figure 4). The PCA plot revealed distinct patterns in the plasma biomarkers of the non-KTRs over the clinical course of the disease (D0, D3, and D7), compared with the HCs (Figure 4A). In particular, D0 exhibited a different pattern compared with D3 and D7, which showed a similarity to the majority of the cases. After hospital discharge (CS30), the plasma biomarkers exhibited a pattern remarkably similar to that of the HCs. The top three biomarkers with the highest potential to differentiate between COVID-19 and the HCs were IL-10, IFN-α, and TNF-α (Figure 4B). 

For the COVID-19 KTRs, distinct patterns were observed in the D0, D3, and D7 groups when compared to the KTR controls, whereas CS30 displayed a highly similar pattern. Furthermore, D0 exhibited a divergent pattern, compared to D3 and D7 (Figure 4C). The top three primary biomarkers with the highest discriminatory potential between the COVID-19 KTRs and their controls were identified as NGAL, sVCAM-1, and IL-8 (Figure 4D).

## 4. Discussion

Early in the pandemic, the cytokine signature triggered by the SARS-CoV-2 infection was recognized as a cytokine storm, providing opportunities for identifying predictors of clinical deterioration and new targets for clinical interventions [11,12,13,14,15]. In this study, we investigated the association of the circulating levels of several mediators in moderately ill COVID-19 patients with their clinical outcomes, including a subset of vulnerable individuals, KTRs. Additionally, we investigated the crosstalk between these mediators, seeking response patterns based on cytokine network correlations.

A significant increase in multiple cytokines and vascular mediators was observed in the COVID-19 patients, regardless of their KTR status. IL-6, IL-8, and IL-10 were higher in the patients with COVID-19 than in their respective controls from baseline to day 7. IL-6 has been considered a pivotal mediator in patients with COVID-19. A meta-analysis that included 10 studies (N = 1789) found 3.24-fold higher levels of IL-6 in patients who were admitted into the ICU, compared with non-severe patients [28]. IL-6, as a marker of inflammation, has been linked with proteome changes consistent with an exacerbation of the acute phase response in COVID-19 patients [29] and was successfully used as an adjunctive target therapy in critically ill patients [30].

It is not surprising that adhesion molecules, such as P-selectin, sICAM-1, and sVCAM-1, differentiated the non-KTR patients from their healthy controls. These molecules mediate leukocytes’ and platelets’ interactions with endothelial cells, eliciting a microvascular response with impaired endothelium-dependent vasorelaxation in the arterioles, excess fluid filtration in the capillaries, and protein extravasation in the venules, resulting in tissue and organ function impairment [31,32]. Another interesting finding was the sustained GDF-15 levels from baseline to day 7, which can be attributed to disease progression, as GDF-15 suppresses antigen presentation via the dendritic cells and is upregulated by IL-1β, TNF-α, IL-2, and macrophage colony-stimulating factor 1 in a complex and tissue-specific regulation pathway [33,34]. On the other hand, GDF-15 modulates vascular responses involving the nitric oxide pathway, impairing the vascular contractile, relaxing functions, and amplifying organ dysfunction [35]. Quite surprisingly, only sVCAM-1 was increased in the KTR patients, compared to their respective controls.

Although the KTR patients with COVID-19 had worse outcomes, the cytokine and vascular mediator levels were not significantly different between the KTR and non-KTR patients, except for NGAL. Unsurprisingly, NGAL differentiated the KTRs with COVID-19 from their respective controls because the KTRs usually progress to acute kidney injury (AKI) during the infection [9,10]. The baseline graft function is associated with COVID-19-associated outcomes among KTRs. For instance, in the TANGO international consortium, which comprised data from KTRs in several countries, the estimated glomerular filtration rate (eGFR) among KTRs requiring hospitalization was lower than 50 mL/min/1.73 m^2^, and a lower graft function differentiated survivors from non-survivors [9]. Similar results were observed in the Brazilian multicenter registry, where the frequency of AKI in KTRs with COVID-19 was 23.2%, and the 90-day case fatality rate was significantly higher in patients with AKI (36.0 vs. 19.1%) and in those who required renal replacement therapy (RRT) during hospitalization (70.8 vs. 10.1%) [10]. Investigating non-transplanted patients presenting in the emergency room with laboratory-confirmed COVID-19, of whom one-third developed AKI, the levels of NGAL showed a good performance for predicting AKI (AUC = 0.81) and a need for RRT (AUC = 0.87) [36]. Most recently, a prospective study investigated the performance of urinary biomarkers in predicting the COVID-19-associated composite endpoint of intensive care unit requirements, including a need for RRT, mechanical ventilation, and in-hospital mortality, and found that urinary NGAL tended to predict AKI (AUC = 0.669, and *p* = 0.05) and in-hospital mortality (AUC = 0.674, and *p* = 0.07) [37]. Besides being a useful marker of kidney injury, it is worthy to emphasize the role of NGAL in the inflammatory response. NGAL is synthesized as a component of the late granules of neutrophils, where it colocalizes with MPO, providing protection against bacterial infection by interacting with bacterial proteins, termed siderophores [38]. It is induced by cytokines and is involved in the host response in different infection models. Although neutrophils are a primary source of NGAL, their expression is also found in numerous human tissues, including the tubular cells in the kidney, heart, and lungs, among others [39]. In sepsis, the serum levels of NGAL were reported to be higher in the patients with a more severe disease, and in agreement with our correlation analysis, NGAL showed significant relationships with inflammatory cytokines and vascular mediators [40].

These punctual differences in cytokines and vascular mediators observed in the patients, regardless of being KTRs or not, were less evident when the levels were stratified by their clinical evolution: critical vs. non-critical. At baseline and on day 3, among the non-KTRs, more mediators presented significant differences in their levels according to clinical progression than those among the KTRs. It is well known that the levels of several cytokines can predict the clinical course, severity, and prognosis of different inflammatory diseases, mainly in the acute phase of sepsis [41,42,43,44]. Generally, these associations are based on mediator levels assessed at determined time points; however, a bivariate analysis in our study could not have captured the differences among the KTRs, in part because of the low number of patients included.

The immune response is a highly coordinated process involving the cross-regulation of cytokines and vascular mediators. The complex interplay between these markers can be investigated by analyzing the correlation matrices commonly employed in numerous clinical scenarios [45,46,47,48,49,50]. Thus, more recently, the visualization of cytokine networks and their association with clinical outcomes have emerged as a milestone in investigating some critical processes, better explaining inflammatory and endothelial dysfunction and consequent organ failure [51,52,53]. In the context of COVID-19, cytokine network analyses have previously demonstrated the association between immune mediators and clinical parameters, such as pneumonia, hypoxia, and ICU admission [53], and the similarities between the responses to influenza and SARS-CoV-2 infections [51,52,53]. Thus far, in our study, we employed the network analysis of the Spearman correlation matrices to examine the correlation profiles of the biomarkers in KTRs and non-KTRs. Our findings revealed significant differences in the correlation profiles between these two groups, encompassing variations in the number of correlations, the directions of the correlations, and the nodes with the highest connectivity. The network of the KTRs showed less interaction (14 vs. 44), and it is noteworthy that the lower density of interactions displayed in the network analysis also points to a possible derangement of the inflammatory response [45,46,47,48,49,50]. This finding suggests a potential disruption or dysfunction in the immune response of KTRs. The node analysis indicated that NGAL and sVCAM-1 exhibited the highest number of interactions in both the non-KTR and KTR networks. These results highlight the possible central roles of these markers in the immune response to SARS-CoV-2 infection, independent of the kidney transplant condition. 

Lastly, the PCA showed distinct patterns between the non-KTRs and KTRs at three different time points during hospitalization and 33 days after discharge, which underscored differences over this time. The PCA gave insight into the dynamics of the altered biomarkers during the acute phase of the disease and into convalescence [54,55]. In both groups of patients, there was a transition between admission to convalescent samples, which showed a similar pattern with their respective controls. This is interesting, considering that the KTRs presented a more complicated clinical course, longer hospital stays, and a higher proportion of patients progressing to critical illness and death, compared with the non-KTRs. However, the top three mediators showing the highest potential to differentiate between the patients’ longitudinal samples and their respective controls were diverse in the KTRs and non-KTRs: IL-8, sVCAM-1, and NGAL for the KTRs, which are involved in chemotaxis, cell–endothelium interaction, and neutrophils’ anti-microbial activity, and IL-10, IFN-α, and TNF-α for the non-KTRs, implying that an inflammatory response is a major factor for the PCA pattern.

Beyond the pattern of cytokines and vascular mediators’ response to COVID-19, three central circumstances differentiate KTRs from non-transplanted patients: their cumulative number of comorbidities, the unavoidable use of anti-rejection immunosuppressive drugs, and the presence of the graft per se. Since the first reports, clinical presentations and outcomes in KTRs have awakened particular concerns, due to their cumulative number of comorbidities [7,8], and a high vulnerability was demonstrated in different cohorts, resulting in high hospitalization rates and impressive case fatality rates [9,10]. However, when KTRs were compared with patients with chronic kidney disease undergoing dialysis with several comorbidities as well, the risk of death was significantly higher for the KTRs [56]. In addition, in a population-based study comparing the COVID-19-associated case fatality rate in the general population with that of KTRs, the overall fatality rate was 24.9% for the KTRs (vs. 3.5% for the general population), with a higher risk across all age groups [6]. In patients aged 20–39 years old with a lower prevalence of comorbidities, the case fatality rate was 23 times higher (8.5 vs. 0.37%) among the KTRs [6]. Both results underscored that prolonged immunosuppression plays a determining role in clinical results, which could be associated with differences in the immune and endothelial responses. Here, we present data showing the similarities and discrepancies in cytokines and vascular mediators between KTR and non-KTR COVID-19 patients.

Our study had some limitations, such as the small number of patients enrolled and the fact that the patients were enrolled at a single center. In addition, we did not investigate the influence of the viral load or the roles of virus variants in the mediators and outcomes. Furthermore, cytokines previously described to play a role in COVID-19, including IL-1β and IL-17A, did not yield consistent results in our study. On the other hand, it had some strengths that can be highlighted, such as its prospective design, with the sequential cytokine and vascular mediator measurements, and the inclusion of a convalescent sample for the survivors as well.

## 5. Conclusions

In conclusion, this study provides significant insights into the inflammatory response in COVID-19 patients, with a particular focus on KTR patients. Despite experiencing worse clinical outcomes, the majority of the cytokine and vascular mediator levels showed no significant differences, compared to those of the non-KTR patients. As an exception, the significantly elevated levels of NGAL in the KTRs suggest its well-established role in kidney injury within this population. The network analysis, however, evidenced differences between the KTR and non-KTR patients, the lower density of interactions suggested potential immune derangement in the KTRs, and the PCA analysis showed different top mediators with the highest potential to differentiate the COVID-19 patients from their respective controls. This study’s findings advance our understanding of the COVID-19 immune response, particularly for vulnerable populations like KTR patients. 

## Figures and Tables

**Figure 1 viruses-15-02166-f001:**
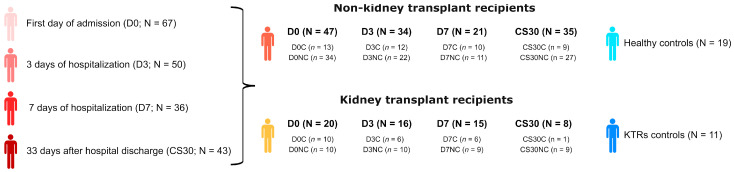
Study design. Patients with COVID-19 were prospectively enrolled, including KTR and non-KTR patients. Healthy volunteers and KTR patients without COVID-19 were included as control groups.

**Figure 2 viruses-15-02166-f002:**
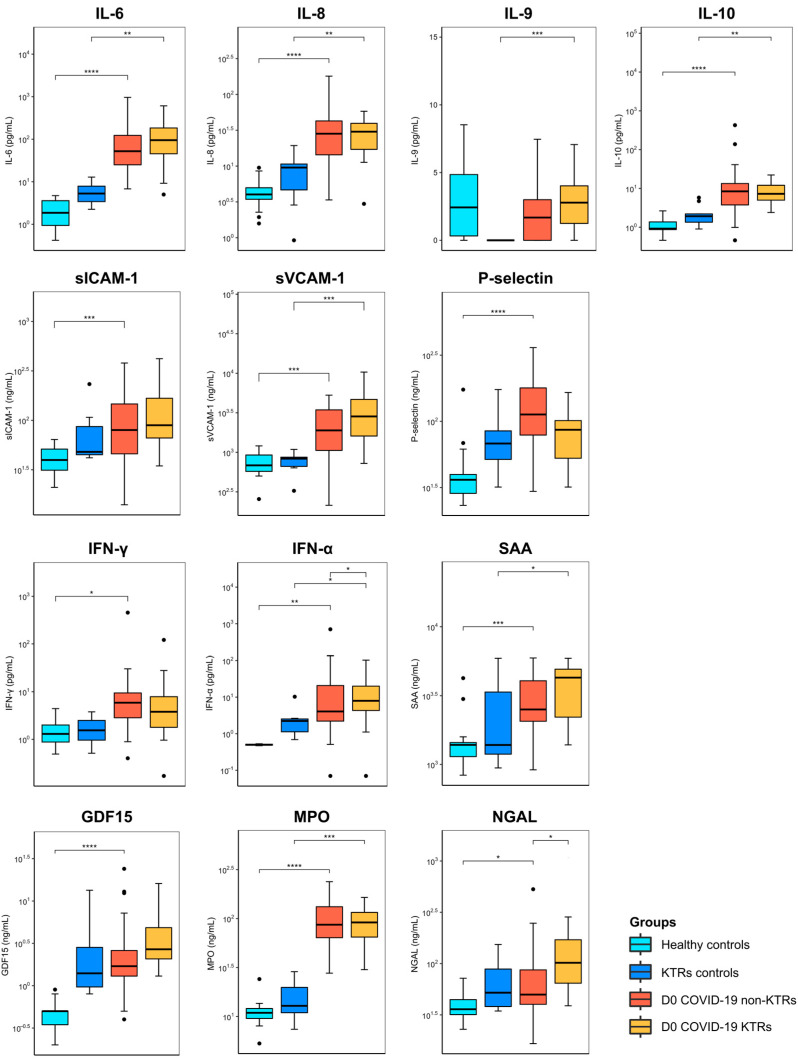
Baseline plasma biomarker levels in patients with COVID-19 stratified by kidney transplant recipient status and respective controls. To investigate potential differences in plasma biomarkers, we employed the Kruskal–Wallis test, followed by Dunn’s test, with Benjamini–Hochberg correction for multiple comparisons. Statistically significant results are denoted by the following symbols: * *p* < 0.05, ** *p* < 0.01, *** *p* < 0.001, and **** *p* < 0.0001. The outliers are showed as black dots. Abbreviations: IL-6: interleukin 6. IL-8: interleukin 8. IL-9: interleukin 9. IL-10: interleukin 10. sICAM-1: soluble intercellular adhesion molecule 1. sVCAM-1: soluble circulating vascular cell adhesion molecule 1. IFN-γ: interferon gamma. IFN-α: interferon alpha. SAA: serum amyloid A. GDF-15: growth/differentiation factor 15. MPO: myeloperoxidase. NGAL: neutrophil gelatinase-associated lipocalin. D0 (day 0): blood samples were collected at admission. KTRs: kidney transplant recipients. Non-KTRs: non-kidney transplant recipients.

**Figure 3 viruses-15-02166-f003:**
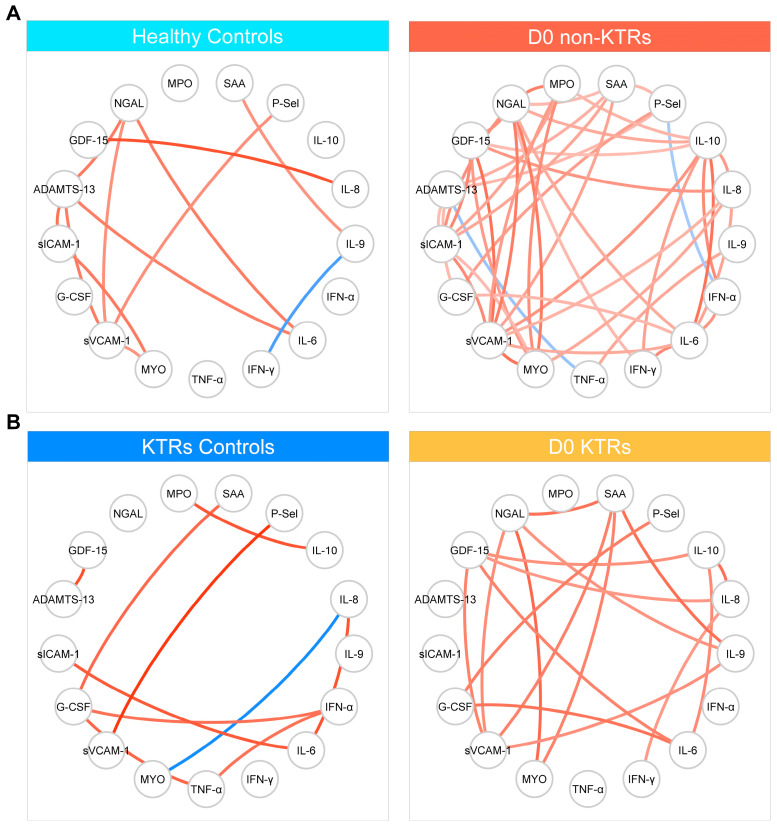
Correlation-based network analysis. (**A**) Non-kidney transplant recipient networks and (**B**) kidney transplant recipient networks. The links displayed in these networks indicate statistically significant Spearman’s rank correlations (*p* < 0.05). Specifically, red links represent positive correlations (rho ≥ 0.20), while blue links represent negative correlations (rho ≤ 0.20). Abbreviations: IL-6: interleukin 6. IL-8: interleukin 8. IL-9: interleukin 9. IL-10: interleukin 10. G-CSF: granulocyte colony-stimulating factor. TNF-α: tumor necrosis factor alpha. IFN-α: interferon alpha. IFN-γ: interferon gamma. GDF-15: growth/differentiation factor 15. sICAM-1: soluble intercellular adhesion molecule 1. MPO: myeloperoxidase. NGAL: neutrophil gelatinase-associated lipocalin. sVCAM-1: soluble circulating vascular cell adhesion molecule 1. SAA: serum amyloid A. MYO: myoglobin. P-Sel: P-selectin. D0 (day 0): blood samples were collected at admission. KTRs: kidney transplant recipients. Non-KTRs: non-kidney transplant recipients.

**Figure 4 viruses-15-02166-f004:**
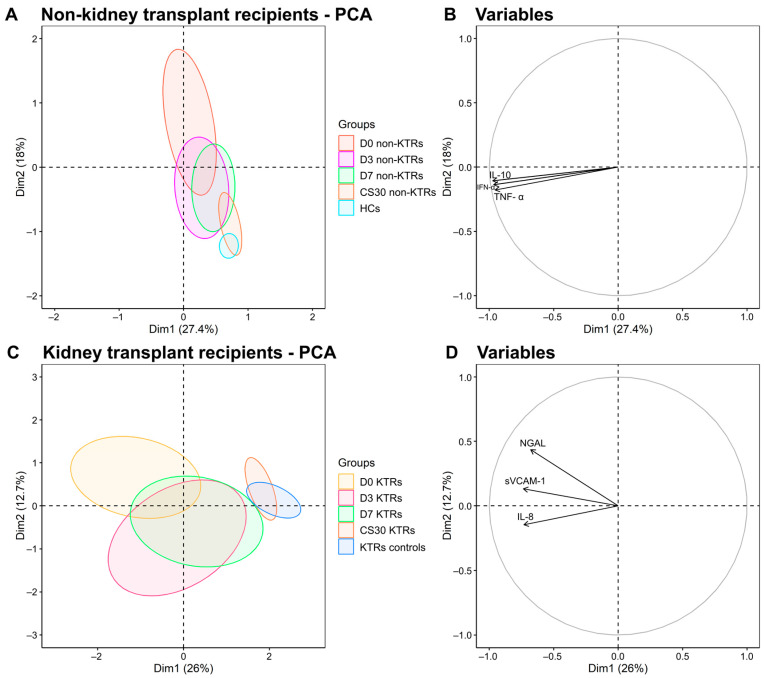
Principal component analysis (PCA) of plasma biomarkers. (**A**) Non-kidney transplant recipient PCA: percentage of explained variance in principal components 1 and 2. (**B**) Contribution of variables (non-kidney transplant recipients): top 3 variables explaining variance in the data. (**C**) Kidney transplant recipient PCA: percentage of explained variance in principal components 1 and 2. (**D**) Contribution of variables (kidney transplant recipients): top 3 variables explaining variance in the data. The ellipse indicates the central 33% of the data points, representing the grouping of the samples. The arrows denote the direction (arrow orientation) and strength (arrow length) of the correlation between each biomarker and the principal components. Abbreviations: IL-10: interleukin 10. IFN-α: interferon alpha. TNF-α: tumor necrosis factor alpha. NGAL: neutrophil gelatinase-associated lipocalin. sVCAM-1: soluble circulating vascular cell adhesion molecule 1. IL-8: interleukin 8. KTRs: kidney transplant recipients. Non-KTRs: non-kidney transplant recipients. HCs: healthy controls.

**Table 1 viruses-15-02166-t001:** The panel of cytokines and proteins related to endothelial cell interactions and inflammation.

Mediators	Methods
IL-6	Cytometric Bead Array (CBA) Flex Set kits
IL-8	The samples were analyzed in LSRFortessa (BD Biosciences, San Jose, CA, USA)
IL-9	
IL-10	
Granulocyte colony-stimulating factor (G-CSF)	
Tumor necrosis factor alpha (TNF-α)	
Interferon alpha (IFN-α)	
Interferon gamma (IFN-γ)	
Disintegrin and metalloproteinase with thrombospondin motifs 13 (ADAMTS-13)	Cytometry (MAGPIX^®^ Instrument, Luminex Corporation, Austin, TX, USA)
Growth/differentiation factor 15 (GDF-15)	Milliplex Map Human Cardiovascular Disease kit Magnetic Bead Panel 2 Cardiovascular Disease Multiplex Assay (Temecula, CA, USA)
Myoglobin (MYO)	
Soluble intercellular adhesion molecule 1 (sICAM-1)	
Myeloperoxidase (MPO)	
P-selectin (P-Sel)	
Neutrophil gelatinase-associated lipocalin (NGAL)	
Soluble circulating vascular cell adhesion molecule 1 (sVCAM-1)	
Serum amyloid A (SAA)	

**Table 2 viruses-15-02166-t002:** Demography, comorbidities, clinical and laboratory admission data, and outcomes stratified by kidney transplant recipient status (yes vs. no).

Variables	KTRs (N = 20)	Non-KTRs (N = 47)	*p*-Value
**Demography**			
Male sex, n (%)	12 (60%)	30 (64%)	0.98
Age, median (IQR)	55.3 (48.9, 60.6)	62.7 (51.2, 67.1)	0.11
**Comorbidities**			
Charlson index, median (IQR)	4.0 (2.0, 5.0)	3.0 (1.0, 5.0)	0.14
Cardiac disease, n (%)	1 (5.0%)	12 (25%)	0.30
Chronic pulmonary disease, n (%)	0 (0%)	9 (19%)	0.08
Diabetes, n (%)	7 (35%)	19 (40%)	0.80
Chronic kidney disease, n (%)	20 (100%)	4 (8.5%)	<0.0001
Hypertension, n (%)	16 (80%)	26 (55%)	0.10
Obesity, n (%)	1 (5.0%)	7 (15%)	0.46
**Admission Data**			
Day of symptoms, median (IQR)	7.0 (5.0, 9.7)	7.0 (5.0, 9.0)	0.93
Fever, n (%)	16 (80%)	31 (66%)	0.30
Cough, n (%)	14 (70%)	35 (74%)	0.90
Shortness of breath, n (%)	15 (75%)	30 (64%)	0.50
Diarrhea, n (%)	7 (35%)	13 (28%)	0.75
Temperature (°C), median (IQR)	36.4 (36.0, 37.2)	36.6 (36.0, 37.8)	0.49
Cardiac rate, bpm	80.0 (78.5, 86.5)	86.0 (78.0, 99.0)	0.23
Respiratory rate, bpm	24.0 (20.0, 26.0)	24.0 (20.0, 26.0)	0.74
SpO2, median of %	91.3 (89.2, 94.1)	90.7 (88.9, 93.0)	0.46
BMI, Kg/m^2^	26.0 (23.4, 27.7)	27.7 (24.4, 32.5)	0.07
SOFA score	2.0 (2.0, 4.0)	1.0 (0.0, 2.2)	0.01
**Laboratory Admission, Median (IQR)**			
Lymphocytes, cells/μL	599.5 (375.2, 906.2)	997.0 (720.0, 1312.0)	0.01
Neutrophils, cells/μL	4647.0 (3918.0, 6901.0)	4668.0 (3577.5, 7072.5)	0.77
Monocytes, cells/μL	407.5 (149.2, 460.8)	294.0 (225.5, 497.0)	0.83
Neutrophil–lymphocyte ratio	7.3 (5.0, 11.6)	4.8 (3.8, 8.7)	0.09
Platelets, cells/μL	190,500 (153,000 238,500)	190,000 (153,500, 233,500)	0.94
Hemoglobin, g/dL	12.4 (11.6, 13.1)	13.4 (12.4, 14.6)	0.02
Hematocrit, median of %	38.1 (35.7, 39.7)	40.0 (36.2, 43.2)	0.08
Creatinine, mg/dL	1.9 (1.4, 2.3)	0.9 (0.7, 1.1)	<0.0001
C-reactive protein, mg/L	103.5 (63.2, 164.5)	81.9 (57.2, 152.1)	0.71
Lactate, mg/dL	14.0 (10.0, 19.0)	12.0 (10.0, 18.0)	0.92
D-dimer, μg/mL	1.0 (0.6, 1.7)	1.2 (0.7, 2.5)	0.48
Troponin, ng/L	19.5 (17.0, 35.0)	8.0 (5.5, 23.0)	**0.03**
**Outcomes**			
Hospital days, median (IQR)	14.5 (9.8, 26.0)	8.0 (5.0, 15.0)	0.02
Severe, n (%)	10 (50%)	13 (28%)	0.10
Mortality, n (%)	8 (40%)	4 (8.5%)	0.01

Abbreviations: KTRs: kidney transplant recipients. Non-KTRs: non-kidney transplant recipients. IQR: interquartile range.

## Data Availability

Data for this manuscript can be made available upon reasonable request to the corresponding author.

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
