# Peer review of "Patterns of Circulating Cytokines and Vascular Markers’ Response in the Presence of COVID-19 in Kidney Transplant Recipients Compared with Non-Transplanted Patients"

_viruses, 2023, doi:10.3390/v15112166_

Round 1

Reviewer 1 Report

Comments and Suggestions for Authors

The authors performed a study on the patterns of circulating cytokines and vascular markers response in the presence of COVID-19 in kidney transplant recipients compared with non-transplanted patients. They pointed out that significantly elevated levels of NGAL in KTRs suggested its well-established role in kidney injury based on their study. Hopefully this study could be valuable to the administration of COVID19 in KTR patients.

Here are some questions and suggestions.

1) The description of grouping of this study is confusing. What does the authors mean by their statement In addition, 10 KTRs with no symptoms related to COVID-19 and no contact with COVID- 19 were matched in terms of time since transplantation, immunosuppressive therapy, sex, and age as a further control group.  ? Furthermore, there is a discrepancy between the grouping presented in the results and the grouping strategy outlined in the materials and methods section. To rectify this, it is advisable for the authors to provide a comprehensive table delineating the grouping strategy and the corresponding results of their study. I noted that the authors provided a series of tables of their grouping design in the supplementary documents. Could they provide a precise version to clarify the general design of grouping of their study?

2) In figure 1, cytokine levels of the group D0 KTRs should be compared to KTRs controls or D0 non-KTRs. Why the authors frequently compared the group D0 KTRs to healthy controls?

3) In the figure legends for Figures 1 and 2, it is recommended that the authors provide an explanation for the term "D0".

4) Certain cytokines associated with the cytokine storm in COVID-19, such as IFN-β, IL-1, and IL-17, were not encompassed within the scope of this study. It is advisable for the authors to address this limitation in their discussion.

5) The authors pointed out that significantly elevated levels of NGAL in KTRs as a conclusion of their study. In the context of the cytokine storm, NGAL may play a role due to its association with inflammation and immune response. The cytokine storm is characterized by an overactive immune response that leads to the excessive release of pro-inflammatory cytokines, which can result in tissue damage and systemic inflammation. It is advisable for the authors to give more discussion on the roles of NGAL in inflammation.

Reviewer 2 Report

Comments and Suggestions for Authors

The work addresses the possible influence of COVID on kidney transplantation. In reality, these patients are much weaker, particularly due to taking drugs that inhibit the inflammatory and immune systems. Therefore, in my opinion, the main limitations of the study are the small number of patients and the fact that the description of the population does not include the drugs they were taking, which could be a confounding factor.

1 - The introduction is good, but I would like to go into more detail about the effects of COVID and the COVID vaccine on cardiovascular and renal aspects.

2 - The data is very well worked out in terms of the statistical treatment used. The graphs are also very important, and are very clear.

3 – The number of patients for a retrospective study is very low, which could be influencing the results. Wouldn't it be possible to increase the number of patients?

4 The discussion could be more in-depth regarding the effects of COVID on the cardiovascular system and therefore on the renal system.

5 - what are the study's limitations?

Reviewer 3 Report

Comments and Suggestions for Authors

The AA investigated the association of several mediators at different time points in 67 non vaccinated COVID-19 infected patients: 47 non-transplanted and 20 kidneys transplanted (Ktx). Each group was furtherly divided in two subgroups: critical and non-critical. A correlation-based network was also built. No significant difference between groups was found except for increased NGAL levels among Ktx. However, at the network analysis a lower density of interactions was evidenced in Ktx due to the immunosuppression.

Major criticism:

-the paper is missing any demographic table: Table 1 and 2 are not included in the supplementary material: Descriptive tables are not supplementary material.

-No information is given about the immunosuppression therapy that the Ktx were following as well as modification if any and specific treatment for the COVID infection (i.e steroid increase?), being the patients enrolled between May and September 2020 several empiric treatments were tried. No comment about the evolution of the disease: 40% incidence of death among Ktx is one of the highest reported in the literature and make me hypnotise that the definition of “moderately ill patients” (Discussion line 4) should be revised as well as what is reported in paragraph 2.1 of Methods: definition of moderate to severe disease according to WHO. Again, a Table with symptoms and final evolution should have been included.

- The conclusion is not evidencing anything really worth addressing. Instead of focusing on some specific markers of infection severity or with prognostic value, the AA measured really a Cytokines storm that allowed the only conclusion that Ktx patients are fragile, at high risk of unfavourable outcome have a lot of comorbid condition influencing the prognosis and do not have a normal kidney function therefore are at risk of AKI that worsen the prognosis “per se”.

Minor criticism

-       Introduction is too long and not completely relevant to the topic of the paper.

-       The paper is difficult to read: somehow confused particularly the paragraph 2.1 in Methods

Comments on the Quality of English Language

Some paragraph are not clear mainly because the phrases are too long (Method and Discussion)

Round 2

Reviewer 2 Report

Comments and Suggestions for Authors

 Accept in present form.

Author Response

Thank you very much for taking the time to review this manuscript. 

Reviewer 3 Report

Comments and Suggestions for Authors The paper has improved noticebly after the Reviewers comments. And although I still believe that it has a very limited clinical interest, I leave to the Editor the final decision about publication.

Comments on the Quality of English Language

none

Author Response

(The authors gave the same response as above.)
